# Dietary Inflammatory Index and Associations with Sarcopenia Symptomology in Community-Dwelling Older Adults

**DOI:** 10.3390/nu14245319

**Published:** 2022-12-15

**Authors:** Corey Linton, Hattie H. Wright, Daniel P. Wadsworth, Mia A. Schaumberg

**Affiliations:** 1School of Health and Behavioural Sciences, University of the Sunshine Coast, Sippy Downs, QLD 4556, Australia; 2Sunshine Coast Health Institute, Birtinya, QLD 4575, Australia; 3School of Nursing, Midwifery and Paramedicine, University of the Sunshine Coast, Sippy Downs, QLD 4556, Australia; 4Manna Institute, University of the Sunshine Coast, Sippy Downs, QLD 4553, Australia; 5School of Human Movement and Nutrition Sciences, the University of Queensland, Brisbane, QLD 4072, Australia

**Keywords:** diet, inflammation, ageing, physical function, muscle strength, muscle mass

## Abstract

Low-grade systemic inflammation is a key driver of muscle degeneration in older adults, and diets with pro-inflammatory properties may further contribute to loss of muscle mass, strength and function. Therefore, this research aimed to explore the associations between the inflammatory potential of the diet and measures of sarcopenia symptomology in community-dwelling older adults. Upper (handgrip strength, HGS) and lower extremity (sit-to-stand) muscle strength, physical performance (timed-up-and-go, TUG) and appendicular skeletal muscle mass (ASM) was assessed according to the European Working Group on Sarcopenia in Older People version 2 (EWGSOP2) criteria. Multiple 24-hr dietary recalls were used to calculate the Dietary Inflammatory Index (DII), which was then used to group participants into anti- and pro-inflammatory dietary groups. Multiple linear regression investigated associations between DII, muscle strength, physical performance, and muscle quantity adjusted for age, gender, comorbidities, waist circumference and physical activity. Adults 65–85 years (n = 110, 72.1 ± 4.7 years, 76.4% female) were recruited. One participant was identified with sarcopenia, 35.2% were pre-frail, or frail. More participants with a pro-inflammatory DII score had low muscle quantity than those with anti-inflammatory DII (3.4% vs. 6.4%, x^2^ = 4.537, *p* = 0.043) and DII was negatively associated with HGS (β = −0.157, *p* = 0.016) and ASM (β = −0.176, *p* = 0.002) which remained significant after adjusting for covariates. In this population, DII was associated with less favorable muscle strength, physical performance, and muscle quantity.

## 1. Introduction

Globally, the population is ageing [1], and with a greater risk of chronic disease and disease-related burden with age [2], it is widely accepted that prevention is better than cure. Thus, there is an increased focus on healthy ageing and maintaining the health of older adults [3]. Sarcopenia is a highly prevalent, age-related chronic disease of the musculoskeletal system and is estimated to affect 10% of older adults globally [4]. Sarcopenia is categorised by the atrophy of type II muscle fibres [5], and a resulting loss of skeletal muscle mass and strength. This is associated with loss of physical function and capacity to complete activities of daily living, and many related detrimental health outcomes such as an increased risk of falls, leading to fractures and frailty, placing immense strain on the health care system [5]. Skeletal muscle is essential for maintaining tissue structure, creating muscular contraction and force transmission; all of which are processes relied upon for healthy ageing [6]. Preserving muscle is not only vital for maintaining independence, but also for its metabolic and homeostatic roles [7]. A decline in muscle mass and strength in older age has been shown to negatively affect the outcomes of several chronic diseases which leads to an increase in hospital admissions and mortality [7]. Although there is no cure, interventions to prevent sarcopenia have been explored, with a focus on maintaining both muscle mass and function to prevent progression of the disease [8]. There are a number of contributing factors to the development of sarcopenia, one of which includes inflammation [9]. Diet is recognized as a contributing factor for inflammation and in turn, healthy ageing [10]. A large body of evidence is available on single nutrients and their influence on inflammatory cytokines as well as their role in healthy ageing [11,12,13]. Key nutrients identified to play an important role in ageing well include protein, vitamin D, calcium, antioxidants and omega-3 fatty acids [12]. Over the past decade it has been recognized that the synergistic effect of nutrients, non-nutrient components, and foods, have beneficial impacts on health outcomes. This has resulted in a paradigm shift away from single nutrient research to dietary patterns to examine the effect of the whole diet [14]. 

Research has explored the role of the Mediterranean diet, coined as an anti-inflammatory dietary pattern, in the prevention of several chronic diseases [15,16,17,18]. The Mediterranean diet is commonly associated with weight loss, reduced chronic inflammation, improved diabetes management and lower cardiovascular disease (CVD) risk [15,16,17,18,19,20]. Emerging research demonstrates the beneficial effect of a Mediterranean diet on body composition changes and functional disability in older adults and decreased risk of sarcopenia [19,20,21,22]. A beneficial association has also been reported between Mediterranean diet adherence and sarcopenia symptomology [22,23]. These findings suggest following a Mediterranean diet can improve sarcopenia symptomology. However, studies exploring the specific mechanism behind this improvement, hypothesized to be the anti-inflammatory effect of the Mediterranean diet is scarce [24,25,26,27,28]. Increased inflammatory biomarkers are implicated in the molecular pathway of age-related skeletal muscle breakdown, contributing to sarcopenia [29]. Therefore, a diet resulting in an anti-inflammatory response may mitigate age-related inflammation and combat skeletal muscle breakdown, thereby improving sarcopenia symptomology.

The dietary inflammatory index (DII) was developed to quantify the inflammatory potential of the diet [10]. Foods such as fruits, vegetables, nuts, seeds, and whole grains are classed as anti-inflammatory foods due to their ability to reduce inflammatory biomarkers whereas, processed foods and animal products is associated with higher circulating concentrations of C-reactive protein (CRP) and Interleukin-6 (IL-6) [30,31,32]. The DII scoring algorithm was developed by collating data from peer-reviewed research articles that reported the effect of specific foods and nutrients on circulating inflammatory biomarkers [10]. The overall score from the algorithm combines the effect of dietary components on inflammation and indicates an overall pro- or anti-inflammatory dietary pattern [10]. Subsequent findings suggest that DII has a direct relationship with circulating markers of chronic inflammation [31]. To date, most research has focused on the association between the inflammatory potential of the diet and chronic disease such as cancer, depression, and cardiovascular disease [24,25,26,27,28]. Despite the strong evidence to support dietary influence on inflammation and the influence of inflammation on the progression of sarcopenia, there are only a handful of studies exploring the association between dietary inflammatory potential *per se* and sarcopenia symptomology, all offering varying results [33,34,35,36,37]. Therefore, this study aimed to explore the associations between the inflammatory potential of the diet and measures of sarcopenia symptomology in community-dwelling older adults.

## 2. Materials and Methods

### 2.1. Participants

Functionally able, community-dwelling older adults aged 65–85 years were recruited as part of a wider community-based evaluation project in partnership with the local government. Participants were recruited from community-based exercise classes, email lists, web-based advertising, presentations and by word-of-mouth. Participants were classified as ‘not high risk of experiencing a cardiac event during exercise’ according to the adult pre-exercise screening system [38] and those with poorly controlled hypertension, cardiomyopathy, unstable angina, heart failure or severe arrhythmia, cancer, or chronic communicable infectious diseases, were excluded. The study was granted ethics approval by the University of the Sunshine Coast Human Research Ethics Committee (#A201498).

### 2.2. Assessment of Dietary Inflammatory Index

Three, 24-hour dietary recalls (24 h recall) were collected from each participant by an accredited practicing dietitian via phone or in person over a two-week period following the multiple pass method [39]. The average of the 24 h recalls represented habitual dietary intake. Recalls were manually entered into FoodWorks^®^ Professional Version 10, an Australian nutrient analysis software program. Energy [40] and nutrient analyses were exported to Excel to use the DII tool. Over- and under-reporting was assessed according to the Goldberg ratio method (reported energy intake to basal metabolic rate, EI:BMR) [41,42]. Cut-points included EI:BMR > 2.62 for men and >2.42 for women as over-reporting and <1.21 for men and <1.11 for women as under-reporting [41]. Basal metabolic rate was calculated using the Mifflin St Joer equation [43]. Both under-and over-reporters were excluded from dietary analysis.

#### Calculation of Diet Inflammatory Index Scores

Dietary data from FoodWorks^®^ was imputed into the developed DII calculation tool to determine the inflammatory potential of the diet as previously described [10,44]. The following food parameters were not included in calculating the DII score due to the difficulty in quantifying portion sizes: garlic, ginger, saffron, turmeric, pepper, thyme/oregano and rosemary. In addition, items not calculated by FoodWorks^®^ namely vitamin D, flavan-3-ol, flavones, flavanols, flavanones, anthocyanidins and isoflavones were not included. Others have also excluded similar foods and non-nutrients from calculated DII scores due to quantification difficulty [33,35,36]. The DII score was used to group participants into either an anti-inflammatory dietary index (DII score < 0) or pro-inflammatory dietary index (DII score > 0) group. 

### 2.3. Assessment of Anthropometry 

Height (Harpeden Wall Mounted Stadiometer), body mass (A&D HW-200KGL Calibrated Scales), and waist circumference (Cescorf Tape Meausre) were measured following standard International Society for the Advancement of Kinanthropometry (ISAK) procedures. Body mass index (BMI) was calculated using the body mass relative to squared height [45]. Underweight (BMI: <24 kg/m^2^), normal weight (BMI: 24–30 kg/m^2^) and overweight (BMI: >30 kg/m^2^) was determined using standard BMI cut-offs for adults over the age of 65-years [46]. Waist circumference identified abdominal obesity as per established criteria for metabolic risk (men: ≥94 cm; women: ≥80 cm) [47].

### 2.4. Assessment of Sarcopenia Symptomology and Functional Frailty

Sarcopenia symptomology was assessed as per the European Working Group on Sarcopenia in Older People version 2 (EWGSOP2) diagnostic criteria which included an assessment of muscle quantity, muscle strength and physical performance [48]. Inter- and intra-reliability testing was undertaken for hand grip strength, timed up and go test, gait speed analysis and all anthropometric measures by the principal investigator and all researchers collecting data prior to data collection, with acceptable variability being <2% for anthropometric measures and <5% for physical performance measures. 

#### 2.4.1. Muscle Quantity

Appendicular skeletal body mass was determine using dual-energy x-ray absorptiometry (DXA, GE Lunar iDXA, GE encore software version 13.60) in accordance to the manufacturer’s instructions and in accordance to the protocol developed by Nana and colleagues [49]. Whole-body DXA scans were conducted between 6–8 am in a sub-sample of participants. Participants were over-night fasted, rested and euhydrated prior to the scan. All participants removed metal objects and jewellery from their body and wore minimal clothing. Velcro straps were positioned around the ankles and torso/arms of each participant to minimise movement during the scans [49]. All scans were conducted by the same Queensland Radiation Health licenced researcher using the predetermined mode by the auto scan feature in the software. Coefficients of variants (CV) for the lab have been reported elsewhere [50] reporting, a CV for lean mass of 2.6%.

Scans were used to determine total body, appendicular and lower limb skeletal muscle mass. Appendicular skeletal muscle (ASM) mass and ASM relative to height (ASMI, ASM/height^2^) identified low muscle quantity (ASMI cut-point for men: <7.0 kg/m^2^ and women: <5.5 kg/m^2^, ASM cut-points for men: <20 kg and women: <15 kg) [48].

#### 2.4.2. Muscle Strength

Upper extremity muscle strength was measured using isometric hand grip strength (HGS) measured using a spring-loaded grip dynamometer (TTM, Tokyo, Japan) with the best of three attempts recorded from the dominant hand [51]. Pre-identified cut-points identified low muscle strength (HGS cut-off men: <27 kg; women: <16 kg) as a measure of sarcopenia symptomology [48,52].

Functional leg strength was assessed using the sit-to-stand (STS) test from a standardised 43 cm chair height. Participants were instructed cross their arms across their chest and rise as fast as possible to a full standing position then return to a full sitting position without using their arms as many times as possible in 30 s [53].

#### 2.4.3. Physical Performance

The timed-up-and-go test [54] was used to assess physical performance. Starting in a seated position, the time it took for participants to rise, walk three meters forward, turn around, walk back to the chair and sit down was recorded. Globally identified cut-points were utilised to identify poor physical performance (TUG cut-point: ≥20 s) [54].

### 2.5. Socio-Demographic and Physical Activity

Participants completed an online survey gathering socio-demographic information including age, gender, marital status, and income, administered via Qualtrics^®^. Leisure time physical activity frequency was assessed with the Godin Leisure-Time questionnaire [55].

### 2.6. Statistical Analysis

Normality of data distribution was assessed via measures of skewness and kurtosis. All continuous variables are expressed as means ± standard deviation and categorical data are expressed as frequencies and percentages. Participants were grouped based on DII score into pro- or anti-inflammatory dietary groups. Independent samples *t*-test were used to compare participant characteristic data and sarcopenia symptomology (muscle quantity, muscle strength and muscle performance) between DII groups. Pearson correlation coefficients (or Spearman’s rho for non-parametric data) were used to explore a correlation between DII, ASM, HGS, STS, and TUG. Multiple linear regression analyses were used to explore associations between DII and sarcopenia symptomology controlling for the following covariates: age, gender, comorbidities, waist circumference and physical activity levels. All data was analysed using Microsoft Excel 2019 and SPSS (version 22.0, SPSS, Inc., Chicago, IL, USA) with significance set at *p* < 0.05.

## 3. Results

A total of 173 participants were recruited for this cross-sectional study, however, after excluding under- (n = 33) and over-reporters (n = 1) and those completing only one 24 h recall (n = 29) the final analysis included n = 110 community dwelling older adults. Participant characteristics across both pro- and anti- inflammatory diets categorised by DII score are presented in Table 1. The average DII score was −0.44 ± 1.64. Participants grouped in the anti-inflammatory dietary inflammatory index group (n = 63, 57.3%) had an average DII score of −1.58 ± 1.08, whereas those in the pro-inflammatory dietary inflammatory index group (n = 47, 42.7%) had an average DII score of 1.09 ± 0.82. Of all participants, 50.9% (n = 56) had a waist circumference above cut points for abdominal obesity [47], whilst 14.3% (n = 16) were identified as overweight or obese (BMI > 30 kg/m^2^) and 36.6% (n = 41) as underweight (BMI < 24 kg/m^2^). The DII scores ranged from −4.43 to 2.96. Sarcopenia was identified in one participant and the majority of participants were retired or received aged pension (84.4%, n = 76) with no differences in participant characteristics, and sarcopenia symptomology with the exception of HGS (*p* = 0.009).

The percentage of participants that were identified with low muscle strength, low muscle quantity or low physical performance as per sarcopenia cut-points grouped according to DII scores are shown in Table 2. A small sample size of participants overall were identified with low muscle quantity (n = 10), more participants in the pro-inflammatory dietary index group had lower muscle quantity (*p* = 0.043) compared to those in the anti-inflammatory dietary index group.

Table 3 presents unadjusted and adjusted models exploring the association between DII and sarcopenia symptomology using multiple linear regression coefficients. The DII was inversely associated with HGS and ASM, as well as positively associated with TUG. Associations remained significant between DII and HGS as well as ASM when adjusting for age, gender and waist circumference in the best fit model (Table 3, Model 3).

## 4. Discussion

The key finding of this study was that a lower DII was associated with higher muscle mass and higher upper extremity muscle strength in functionally able community dwelling older adults.

The existing body of evidence that explored the associations between DII and sarcopenia symptomology is inconclusive. Evidence has shown that higher DII scores are associated with lower muscle mass, slower gait speed and increased risk of sarcopenia [33,35,36]. In a cross-sectional study, Gojanovic et al. [35] investigated the associations between the inflammatory potential of the diet, muscle mass and physical performance in 809 older adults (60–95-years). The authors found similar associations between DII and muscle mass (β = −0.13, *p* < 0.001) to the present findings, however they also report a positive association between DII and physical performance assessed by the TUG test (β = 0.02, *p* = 0.035). Laclaustra et al. [36] also reported a favorable association between low DII scores and better physical performance. The authors reported that those with high DII scores were at higher risk of slower gait speed in a cohort of older (>60 years) Spanish adults (OR = 1.82, *p* = 0.001) [36]. The exact reason these findings differ to those presented in the current study is unclear. However, there are several differences which must be noted; both Gojanovic et al. [35] and Laclaustra et al. [36] had larger sample sizes (n = 809 and n = 1948, respectively) than the present study. The smaller sample size of the present study could be why no associations were found between DII and physical performance or potentially due to the current sample being deemed functionally able creating a potentially bias sample. Both studies also used food frequency questionnaires to assess dietary intake. There is known doubt by a large cohort study on the precision of this method for detecting moderate diet–disease associations [57]. At present, the association between DII and physical performance remains inconclusive and more studies are needed to provide clarity regarding these potential relationships.

A longitudinal study [37] exploring the associations between DII and frailty in a population of American adults (aged 45 to 79 years) reported a DII range of −5.65 to +3.70. Although the main outcome was frailty, the authors assessed lower extremity muscle strength through the chair sit to stand test and found no difference in performance between those who followed a pro-inflammatory diet and those who followed an anti-inflammatory diet [37]. Bagheri et al. [33] sought to explore the relationship between dietary inflammatory potential and the risk of sarcopenia in 300 older Iranian adults. They found no association between DII scores and sarcopenia symptomology including muscle strength and muscle mass independently. However, when combined as a complete assessment of sarcopenia the authors reported that a higher DII score was associated with an increased risk of sarcopenia (OR = 2.18, 95%CI = 1.01–4.74). Cervo et al. [34] also explored associations between DII and sarcopenia symptomology in Australian adults aged over 50 years. This study explored the association between DII and musculoskeletal health and reported no significant association with ASM and HGS. Both these outcomes differ from findings in the current study where we report a negative association between DII scores, HGS (β = −0.157, *p* = 0.016) and ASM (β = −0.176, *p* = 0.002). Our finding that HGS was associated with DII even when key participant characteristics were controlled for, is novel and indicate that DII may contribute to poorer upper body muscle strength.

While some studies have reported an association between DII and sarcopenia diagnosis, results are inconclusive in studies independently exploring associations between DII and specific sarcopenia symptomology. This highlights the need for more research to further evaluate whether the inflammatory potential of diet influences the progression and worsening of sarcopenia symptomology [33,35,36]. Chronic low-grade systemic inflammation is known to accelerate muscle loss in older adults [58]. Therefore, an association between the inflammatory potential of the diet and sarcopenia symptomology is plausible and the inconclusive findings in the literature indicate that more research is needed to understand the relationship between DII and sarcopenia symptomology.

This study is one of a handful that have explored DII and its associations with all sarcopenia symptomology to assess muscle strength, muscle quality and physical performance in a large sample of community-dwelling older adults.

The exclusion of several key nutrients used in the DII calculation in the present study may have impacted the overall DII score, however, Hébert et al. [59] found excluding similar dietary parameters from calculations did not reduce the validity of the DII calculation. While 33 participants (19% of the study sample) were excluded from analysis as under-reporters, this appears to be consistent with other studies. A review by Poslusna et al. [60] stated that of published papers collecting dietary data using 24 h recalls, exclusion due to poor data quality such as under-reporting ranged from 21.5–67% of study samples. Therefore, excluding 19% of our sample as a result of under-reporting is consistent with other studies in the field. Further, limitations in the DII tool itself is acknowledged, such as not accounting for supplementation use, and the focus on single foods and nutrients rather than dietary patterns [61]. Whilst these limitations are associated with the tool itself, they must still be acknowledged as a limitation of the present study. The addition of circulating inflammatory biomarker analysis would have enabled us to solidify the influence of dietary inflammation on systemic inflammation to investigate whether this is an important factor influencing the difference in sarcopenia symptomology between groups. While Shivappa et al. [44] previously validated the DII tool as a method to predict circulating CRP concentrations, to-date, blood analysis has not been included in a study exploring the association between DII and sarcopenia symptomology, highlighting an area of focus for future research. Lastly, we did not adjust for energy intake which needs to be considered when interpreting DII scores. Due to small sample size results should be interpreted with care and the nature of a cross-sectional study design precludes conclusions regarding the causality between dietary inflammation and sarcopenia symptomology.

## 5. Conclusions

In conclusion, a diet of low inflammatory potential was associated with greater muscle strength and muscle quantity in this group of functionally able, community dwelling older adults. Following a pro-inflammatory diet was associated with poorer sarcopenia symptomology, which indicates that dietary inflammatory potential may be an important modifiable risk factor for combating the progression of sarcopenia. This is an important finding as it demonstrates the potential impact of the whole-of-diet composition on the prevention of sarcopenia in functionally able, community dwelling older adults. However, further studies, including longitudinal and dietary interventions, are needed to further explore the relationship between dietary inflammation potential, sarcopenia symptomology and functional frailty.

## Figures and Tables

**Table 1 nutrients-14-05319-t001:** Participant characteristics of total group and dietary inflammatory index groups.

Dietary Inflammatory Index Group				
	Total	Anti-Inflammatory (n = 63)	Pro-Inflammatory (n = 47)	*p*-Value ^a^
Participant Characteristics				
Age (years)	72.1 ± 4.7	71.8 ± 4.4	72.4 ± 5.0	0.498
Female n (%)	84 (76.4)	45 (71)	39 (83)	0.180
Weight (kg)	70.6 ± 13.0	71.6 ± 12.6	69.3 ± 13.4	0.373
Height (m)	1.65 ± 0.07	1.67 ± 0.07	1.63 ± 0.07	0.100
Body Mass Index (kg/m^2^)	25.8 ± 4.3	25.7 ± 4.0	25.9 ± 4.7	0.727
Waist circumference (cm)	86.4 ± 12.2	86.3 ± 11.9	86.5 ± 12.6	0.944
2 or more co-morbidities n (%)	18 (16.4)	9 (14.3)	9 (19.1)	0.595
Leisure Time Exercise (n = 110)				0.512
Insufficiently/Moderately Active	10 (9.1%)	7 (6.4%)	3 (2.7%)	
Active	100 (90.9%)	56 (50.9%)	44 (40.0%)	
Marital Status (n = 88):				0.257
Married or Partnered	56 (65.9%)	36 (40.9%)	20 (22.7%)	
Single/Widowed	16 (14.5%)	6 (5.5%)	10 (9.1%)	
Separated or Divorced	16 (18.2%)	8 (9.0%)	8 (9.0%)	
Highest Level of Education (n = 88):				0.272
Primary/Secondary education	6 (5.5%)	5 (4.5%)	1 (0.9%)	
Vocational education	24 (27.2%)	15 (17.0%)	9 (10.2%)	
Tertiary education	58 (68.2%)	30 (34.1%)	28 (31.8%)	
Household Income [56] ^b^ (n = 87):				0.155
Lower income	32 (36.8%)	14 (16.1%)	17 (19.5%)	
Low income	29 (33.3%)	17 (19.5%)	11 (12.6%)	
Middle/High income	16 (18.4%)	10 (11.5%)	6 (6.9%)	
*Undisclosed*	10 (11.5%)	6 (6.9%)	4 (4.6%)	
**Sarcopenia Symptomology**				
Hand grip strength (kg)	27.0 ± 7.6	28.6 ± 8.0	24.8 ± 6.6	0.009
Sit-to-stand test (reps)	14.7 ± 4.6	14.8 ± 4.7	14.5 ± 4.5	0.719
Timed up and go (sec)	5.9 ± 1.1	5.8 ± 0.9	6.0 ± 1.3	0.476
Appendicular Skeletal muscle mass (kg) (n = 87)	18.56 ± 3.84	19.18 ± 3.78	17.59 ± 3.79	0.058
Appendicular Skeletal muscle mass index (kg/m2) (n = 87)	6.7 ± 1.0	6.8 ± 1.0	6.5 ± 1.1	0.224

^a^ *t*-test analysis compared DII groups for quantitative variables and chi-square tests for qualitative variables, Fishers exact has been used for variables where n < 5. ^b^ Australian Dollars.

**Table 2 nutrients-14-05319-t002:** Participants presenting with low muscle strength, quantity and physical performance.

	Dietary Inflammatory Index Group				
Sarcopenia Symptomology ^b^	Total	Anti-Inflammatory (n = 63)	Pro-Inflammatory (n = 47)	x^2^	*p*-Value ^a^
Low muscle strength (n = 110)	7 (6.4%)	2 (1.8%)	5 (4.5%)	2.517	0.135
Low muscle quantity (n = 87)	10 (11.5%)	3 (3.4%)	7 (6.4%)	4.537	0.043
Low performance (n = 110)	0 (0%)	0 (0%)	0 (0%)	-	-

^a^ Obtained from chi-square for qualitative variables, Fishers exact has been used for variables where n < 5 (*p* < 0.05 significant); ^b^ Categorised as per the amended European Working Group on Sarcopenia in Older People diagnosis (EWGSOP2) [48].

**Table 3 nutrients-14-05319-t003:** Multiple linear regression coefficients expressing associations between markers of sarcopenia symptomology and physical frailty and Dietary Inflammatory Index.

Model 3 ^c^				Model 2 ^b^				Model 1 ^a^				Unadjusted Model				
*p*-Value	Β (95%CI)		R^2^	*p*-Value	Β (95%CI)		R^2^	*p*-Value	Β (95%CI)		R^2^	*p*-Value	Β (95%CI)		R^2^	
Hand grip strength																
0.015	−0.160	(−1.303, −0.145)	0.642	0.015	−0.160	(−1.297, −0.146)	0.646	0.016	−0.157	(−1.285, −10.133)	0.645	0.009	−0.249	(−2.017, −0.299)	0.053	DII
Timed up and go																
0.146	0.138	(−0.33, 0.219)	0.234	0.141	0.139	(−0.032, −0.292)	0.24	0.179	0.127	(−0.029, 0.228)	0.233	0.046	0.191	(0.003, 0.251)	0.028	DII
Appendicular skeletal muscle mass (kg)																
0.016	−0.157	(−0.684, −0.162)	0.754	0.002	−0.182	(−0.682, −0.164)	0.757	0.001	−0.206	(−0.759, −0.198)	0.712	0.023	−0.243	(−1.060, −0.080)	0.048	DII

DII, Dietary Inflammatory Index. ^a^ Adjusted for age, gender and number of comorbidities. ^b^ Adjusted for age, gender, waist circumference and number of comorbidities. ^c^ Adjusted for age, gender, waist circumference, number of comorbidities and physical activity levels.

## Data Availability

Data supporting reported results are available upon reasonable request and in accordance with the ethical principles.

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
