# Peer review of "Dietary Inflammatory Index and Associations with Sarcopenia Symptomology in Community-Dwelling Older Adults"

_nutrients, 2022, doi:10.3390/nu14245319_

Round 1

Reviewer 1 Report

Please take a look at the Chi-square result in Table1. Correct the analysis as suggested. Small groups could be collapsed for analysis. e.g leisure time exercise, inactive and moderate could be combined for the analysis, even if you stick to the current data presentation. Otherwise, the chi-square analysis is not valid due to small expected counts. 

Please consider including the confidence interval for the beta (regression coefficient).

Reviewer 2 Report

The manuscript "Dietary inflammatory index and associations with sarcopenia symptomology in community-dwelling older adults" evaluates the associations between dietary inflammatory index and skeletal muscle health. Overall, the data has relative small sample size and is lack of innovation. 

1. Introduction: as the authors stated, there has already been several studies (some were prospective study with large sample size) to investigate the associations between DII and skeletal muscle health. The sample size in the present study was only 110 and the design was cross-sectional, which was lack of innovation.

2. Methods: the DII should be adjusted by total energy intake as nutrients, foods group or dietary patterns are closely related to total energy intake.

3. Methods: Among 173 participants, the percent of under-reporter is (n=33) around 20%, which proved that the quality of the dietary recalls was poor.  

4. Methods: How the authors assess the repeatability of measurements? Were there any data in vivo coefficients of variation of the duplicated measurements of sarcopenia symptomology in this study? 

5. Limitation: I do not agree with the author-stated strength about confounding adjustment, there were limited covariates included, other factors including the demographic status, supplements use and energy intake were not applied for adjustment. 
